# Monitoring Appropriate Monoclonal Antibodies Prescribing via Administrative Data: An Application to Psoriasis

**DOI:** 10.3390/ph15101238

**Published:** 2022-10-08

**Authors:** Elisa Guidotti, Chiara Seghieri, Bruna Vinci, Alice Borghini, Francesco Attanasio

**Affiliations:** 1Management and Health Laboratory, Institute of Management and Department EMbeDS, Scuola Superiore Sant’Anna, 56127 Pisa, Italy; 2SSFO, Pharmacy Department, University of Pisa, 56126 Pisa, Italy; 3Drugs and Appropriateness Policy Sector, Regional Government, 50139 Florence, Italy

**Keywords:** appropriateness, monoclonal antibodies, psoriasis, RWE, administrative data

## Abstract

The Italian Medicines Agency (AIFA) and the Italian Regional Health Systems have implemented measures together with data collection and analysis to improve medicines’ appropriate prescription. Administrative databases represent rich Real-World Evidence (RWE) sources that may be leveraged for research purposes. Thus, such heritage may allow for appropriate prescription studies to be carried out on complex pharmaceutical molecules, as the appropriateness of prescriptions is essential both for patients’ treatment and to ensure healthcare systems’ sustainability. This study analyzed the appropriate monoclonal antibodies (mAbs) prescribed in psoriasis treatment across Tuscany, Italy. Data were extracted from several large administrative databases collected by the Tuscan Regional Healthcare System through record linkages. The analysis showed that over 30% of the 2020 cohort of psoriatic patients could be regarded as potentially inappropriate treated, signaling that mAbs are often prescribed as first-line treatment contrary to guidelines. Variation was observed in the appropriate prescription of mAbs, across different types of mAbs and areas. The study revealed potential inappropriate prescription, and its geographic variation should raise awareness among managers about the appropriate use of resources. Despite limitations, this could represent a pilot for future studies to evaluate the appropriate prescription of mAbs in other clinic conditions and across time.

## 1. Introduction

The appropriate prescription of medication (or appropriate prescribing) is a widely debated issue in literature and has been the target of discussions among national and international Institutions for many years [1]. Barber N. claimed that appropriate prescribing should “maximize efficacy and safety, minimize costs and respect patient’s preferences” [2].

AIFA (*Agenzia Italiana del Farmaco*), the Italian Medicines Agency, defined appropriate prescribing as “the adequacy of the actions adopted to manage a disease, concerning both the patient’s need and the correct use of resources” [3]. The appropriate prescribing of medication and adherence to treatment are particularly relevant both for the efficacy and safety of drug treatments and for the efficient allocation of resources [4]. Indeed, the Italian National Health Service (INHS) has implemented measures to improve appropriate prescribing both at the central and regional levels. More specifically, at the central level, AIFA implemented four measures: the “AIFA notes” that define those therapeutic indications for which the drug is reimbursable by the INHS and encourage physicians to prescribe drugs according to the evidence available in the literature [5]; price caps for single medicines or therapeutic classes negotiated with pharmaceutical companies through management entry agreements [6]; “therapeutic plans” consisting of guidelines for the prescription of a bunch of drugs based on scientific evidence; and “monitoring registries” that collect the flow of treatments for patients who have been prescribed a drug AIFA identified to be monitored for one or more indications through the registries [7]. At the regional level, on the other hand, Regional Health Authorities have implemented multiple measures together with constant data analysis aiming at improving appropriate prescription [8].

Historically, randomized controlled trials (RCTs) have been the main source of data for monitoring and evaluating pharmaceuticals, such as for conducting analysis of prescriptive appropriateness. However, due to several RCT’s limits (i.e., biased populations and lack of generalizability), Real World Evidence (RWE) sources have gained attention over the traditional RCT “gold standard” of evidence. Among others, administrative databases represent rich sources of information that may be leveraged for research purposes [9]. Worldwide, in recent years multiple administrative databases have been used to evaluate the appropriate prescription of medications [10,11,12,13]. AIFA and Italian Regional Healthcare Systems systematically collect and use administrative data to measure the appropriate prescription of medicines through a set of indicators focused on prescription behaviors, consumption of medicines, and compliance to prescribed therapies [3,14]. Thus, the heritage of administrative data may allow for several appropriate prescription studies to be carried out on complex pharmaceutical molecules, as the appropriateness of prescriptions is essential both for patient treatment and to ensure healthcare systems’ sustainability [15].

Thus, the originality of this study consists of the chance of adopting an RWE source, specifically administrative data, to analyze the appropriate monoclonal antibodies (mAbs) prescribed in psoriasis treatment across Tuscany, Italy. Psoriasis is a chronic, genetically influenced, deferring, and relapsing scaly and inflammatory skin disorder that affects 1 to 3 percent of the world’s population. Moreover, psoriasis is a disabling disease with a social and economic impact. Over the past years, therapeutic advances have been made to improve the care of severely affected patients. Indeed, several innovative drugs became available on the market together with the already available long-standing therapies [16]. Among those, mAbs represent powerful human-made treatments. Indeed, mAbs are nowadays used to treat different immunopathological conditions (e.g., psoriasis) since their targets are expressed on the immune system cells [17,18] and appropriate prescription is fundamental to guarantee their proper functioning. Our research is in continuity and evolution compared to a study conducted by AIFA in the period 2014–2015 [19,20].

The aim of the present study is thus twofold. First, to examine the appropriate prescription of a cluster of mAbs (Adalimumab, Certolizumab, Etanercept, Guselkumab, Infliximab, Ixekizumab, Sekukinumab, Ustekinumab, Apremilast, Brodalumab, Risankizumab, Tildrakizumab) in psoriasis treatment in the Tuscany region: as their prescription has been rising over the last few years, thus exerting significant economic pressure on the Tuscan Regional Healthcare System. Second, to verify the presence of intra-regional variation in the prescriptive appropriateness of the mAbs listed above.

## 2. Results

In 2020, 5743 individuals were identified with any type of psoriasis, thus representing the selected cohort. Among those, in the Tuscany region, more than 55% of psoriatic patients were treated with a first-line drug and started the treatment in the same line. Around 11% of those patients received a second-line treatment as the last supply but started with a first-line one. Those were considered appropriately treated according to the study methods. Differently, over 30% of the cohort patients (31.22%) could be regarded as potentially inappropriately treated since they received a second-line treatment as the last supply and started with a second-line one as well. Those patients seemed not to have been supplied with first-line treatment, as recommended by the guidelines. Similarly, potential inappropriateness occurred for the 2.63% of patients that received a second-line treatment as the first supply and the first line as the last supply (see Table 1). Little variation was observed across the different AVs. The percentages of potential inappropriately treated patients were, respectively: 31.66% in North-West AV, 27.57% in Centre AV, and 36.10% in South-East AV. Further details are visible in the Appendix A.

Across the cohort, a total of 1976 patients were treated in 2020 with a group of second-line mAbs, as listed in Table 2. Data for Apremilast, Brodalumab, Risankizumab, and Tildrakizumab were not reported since numbers were particularly limited. Furthermore, a group of patients received the last supply before 2020, and no further reported treatment. Specifically, Adalimumab (11.04%), Etanercept (7.17%), Sekukinumab (4.96%), and Ustekinumab (4.50%) were the most frequently prescribed mAbs for psoriasis in 2020 in Tuscany. Variation occurred across different AVs, specifically for Adalimumab (North-West AV, 13.41%; Centre AV, 8.31%, South-East AV, 11.97%), Sekukinumab (North-West AV, 4.87%; Centre AV, 4.31%, South-East AV, 6.05%) and Ustekinumab (North-West AV, 4.87%; Centre AV, 3.55%, South-East AV, 5.45%). Further details are shown in the Appendix A.

As for the appropriate prescription of mAbs, variation occurred both across mAbs and AVs.

Most of the psoriatic patients treated in 2020 with Adalimumab started the treatment with Adalimumab as well (64.19% in the Tuscany region). Minimum differences were observed across AVs. Most of the patients were thus potentially inappropriately treated since only a fourth (26.02% in the Tuscany region) received it as first supply a first-line treatment.

Similarly, it occurred for Etanercept. Around 30% (Tuscany region) of the psoriatic patients treated in 2020 with Etanercept received a first-line treatment as first supply, thus appropriately treated. Most of the psoriatic patients started the treatment with Etanercept as well (61.89% in the Tuscany region), and no difference occurred across AVs. Those patients were thus potentially inappropriately treated. Figure 1 shows each mAb at the regional level:

–The percentage of 2020 psoriatic patients that received first supply a first-line treatment (i.e., appropriately treated according to the guidelines);–The percentage of 2020 psoriatic patients that received the selected mAb (e.g., Adalimumab) as the first supply is thus considered potentially inappropriately treated.–The percentage of 2020 psoriatic patients that received a mAb (different from the selected, e.g., different from Adalimumab) as the first supply, thus considered potentially inappropriately treated. Complete data, including AV, are visible in the Appendix A.

A different path was observed for Certolizumab, Infiximab, Secukinumab and Ustekinumab. Specifically, most of the psoriatic patients treated in 2020 with one of the above-mentioned mAbs either started the treatment with the same mAb or with second-line treatment, thus mostly accounting for the potentially inappropriate prescription. As an example, among the patients that were treated with Certolizumab in 2020, more than 80% cumulatively were either treated with Certolizumab as a first supply or with a second-line drug. Furthermore, variation in the appropriateness of prescription was observed across AVs for Certolizumab (e.g., South-East AV registered high potential inappropriate prescription). The majority of the psoriatic patients treated in 2020 with Infliximab seemed to have received the same molecule as the first supply. Indeed, the inappropriate prescription was potentially quite relevant. A large proportion of the psoriatic patients treated in 2020 with Sekukinumab mainly started the treatment with a second-line treatment diverse from Sukukinumab, while around 25% (Tuscany region) received Sekukinumab as the first supply. The potentially inappropriate prescription was quite high as well. No difference was observed. A similar trend was observed for Ustekinumab, with remarkable variation. Specifically, South-East AV seemed to differ much from the others. Indeed, it seemed to register lower appropriate prescriptions. Specifically, most of the psoriatic patients treated in 2020 with Ustekinumab either started the treatment with Ustekinumab or with a second-line treatment.

As concern Guselkumab and Ixekizumab, most of the psoriatic patients treated in 2020 with those two mAbs either received a first-line treatment as the first supply or a diverse second-line treatment and then switched to these molecules as the last supply. Specifically, around 40% (Tuscany region) of the psoriatic patients treated in 2020 with Gulsekumab could be regarded as appropriately treated. Little variation occurred across areas. A similar pattern occurred for Ixekizumab. The majority of the psoriatic patients treated in 2020 with Ixekizumab seemed to have received either a first-line treatment as the first supply or a second-line different from Ixekizumab. Indeed, almost half of the cohort seemed to be appropriately treated, while the other appeared potentially inappropriate.

The analysis allowed for conducting some estimation of the costs the Tuscan Regional Healthcare System bears due to potentially inappropriately treated patients. The pharmaceutical databases provided the necessary information for calculating the average cost per supply and the standard deviation (SD) for all the molecules included in the study, as reported in Table 3. Significant variability was observed for Cyclosporin and Infliximab due to the difference in price between the biosimilar medicine and the originator.

Hypothesizing that patients receive four supplies per month and considering the potentially inappropriately treated patients per molecule (Appendix A), benchmarking across treatments’ costs can be carried out. As an example, a month of pharmaceutical treatment with Methotrexate would cost the INHS around €16,369.54, only considering the psoriatic patients treated in 2020 with Adalimumab that received Adalimumab as the first supply. A month of treatment for the same number of patients with Adalimumab would cost €213,813.4. Indeed, the average cost the system sustains per mAb supply is by far notably higher than the amount paid for a first-line drug.

More broadly, considering only those patients that received the selected molecules as first-supply (1004 patients) as shown in the Appendix A, the Tuscan Regional Healthcare System would have spent €40,380.88 for treating all the patients for a month with Methotrexate, €156,403.12 with Cyclosporin, and €36,746.33 with Acitretin. An amount of €2,638,550.94 would have been needed for treating the same cohort for a month with selected mAbs. Indeed, considering a single month of treatment, the region could save €2,598,170.06 (Methotrexate versus mAbs treatment), €2,482,147.82 (Cyclosporin versus mAbs treatment), €2,601,804.54 (Acitretin versus mAbs treatment).

## 3. Discussion

Real-world data are gaining increasing worldwide attention, and their adoption for regulatory, drug development, and healthcare decision-making is rapidly expanding [21]. Indeed, administrative databases represent large sources of data to carry out evaluations on the appropriate prescription of medications. In fact, both AIFA and Regional Health Systems systematically collect and use administrative data to measure the appropriate prescription of medicines, meanwhile conducting ad-hoc analyses. This study focused on the chance of exploiting large regional Tuscan administrative datasets to analyze the appropriate monoclonal antibodies (mAbs) prescribed in psoriasis treatment. The study managed to trace the cohort of psoriatic patients in 2020 and, within the cohort, those who were supplied with mAbs in the same period. The prevalence of psoriasis emerged as lower than expected [22], hypothetically due to the 2020 year (selected as the year of the pandemic year and the impossibility of tracking exemption codes). Furthermore, a retrospective analysis was carried out to catch the line of treatment those received as first supply. The analysis was carried out for each identified mAb in order to identify the level of appropriate prescription of mAbs according to the guidelines.

The analysis shows that over 30% of the cohort patients could be regarded as potentially inappropriate treated, signaling that mAbs are quite often prescribed as first-line treatment in contrast to what is suggested by the drug compendia [19,20], international and national guidelines [23,24,25]. Those results are approximately in line with data reported in the 2014–2015 Osmed Report [19,20], signaling that little improvement has been registered in the 2016–2020 period. Indeed, guidelines suggest prescribing the above-listed mAbs as second-line therapy when the first-line does not work properly. It is largely demonstrated that inappropriate prescribing of medicines can cause adverse drug-related events [26]. Indeed, inappropriate administration of mAbs can lead to several adverse events, including immune reactions (e.g., acute anaphylaxis, serum sickness, the generation of antibodies), infections, cancer, autoimmune disease, and organ-specific damages such as cardiotoxicity [27]. Moreover, prescribing mAbs as the first-line treatment reduces the patient’s therapeutic options in a context where the available therapies are already scarce. Furthermore, in line with previous studies [28,29], mAbs treatments emerged to be more expensive than traditional molecules, thus exerting significant economic pressure on regional healthcare systems. On the other hand, mAbs present some advantages that make them preferable to low-molecular-mass drugs. Among the positive aspects, there are their high specificities associated with more precise action and their long half-lives that allow for reducing dosing frequency [27]. Infrequent dosing may enhance patients’ therapy adherence [30,31].

It is important to highlight that only a limited percentage of the cohort of psoriatic patients is currently treated with mAbs. However, variation was observed in the appropriate prescription of mAbs, both across mAbs and AVs. Specifically, most of the patients treated in 2020 with a second-line mAb either started with a second-line treatment or with the same molecule, despite differences across mAbs. The level of potentially inappropriate prescription is thus particularly high. Moreover, geographical variation in drug prescription is a phenomenon that has been frequently observed in several studies on diverse medicines [32,33,34] and has emerged to have a significant impact on the population’s health [35]. In Tuscany, the level of potentially inappropriate prescription changed among the AV, showing that the probability for a patient to receive an appropriate prescription varies according to the place of care, the providers, and the professional responsible for the patient management. The literature demonstrates how patients who are treated by professionals with different prescribing behavior can have different treatment experiences [36]. Accordingly, geographic variation could lead to equity issues in patients’ treatments. Managing geographic variation thus emerges as fundamental to guaranteeing treatment equity among patients. A crucial role in governing geographical variation is played by Performance Evaluation Systems (PES). PESs provide for systematic benchmarking and public disclosure of data and robust tools to reach health care systems advancement [37]. The Tuscan Regional Healthcare System can take advantage of the Italian Regional Performance Evaluation System (IRPES). The IRPES include a set of indicators dedicated to the monitoring of appropriate drug prescriptions. Among those, a bunch of indicators measures the appropriate prescription of antibiotics, other opioids, antidepressants, and antipsychotics. No indicators are currently available in the system to monitor the appropriate prescription of mAbs. The analysis could thus be a starting point to discuss with the IRPES members the possibility of introducing the mAbs developed indicators in the system. Including the mAbs indicators in the IRPES would allow for continuous and systematic benchmarking among healthcare providers, which has been demonstrated to be a vehicle for continuous performance improvement [38].

Indeed, this study has several limitations. Firstly, the retrospective analysis considered a five-year period because of data availability. Psoriasis, however, is a chronic pathology, and patients could have started their treatment far behind the period considered. Furthermore, 2020 was a pandemic year, and several people did not receive any hospitalizations or visits. Finally, people with hospitalization for psoriasis reasons could have another immune-mediated disease as primary disease and psoriasis as comorbidity or could have received mAbs for reasons other than psoriasis. Secondly, the study was carried out across a single Italian region, without other regional or country comparisons. Thirdly, the study used administrative data as a single source: thus, it includes all the possible biased administrative data generally present (e.g., lack of data because of deficiencies in compilation duties) [39]. Moreover, administrative data do not contain clinical information, making it difficult to get the outcomes produced by potential inappropriate prescription behaviors. Furthermore, drugs switching between the first and last supply were not considered in this analysis, and a simple cost analysis was carried out. However, this study could represent a pilot for future studies to evaluate the appropriate prescription of mAbs in other clinical conditions (e.g., inflammatory bowel disease) and across time. Moreover, the topic of appropriateness should be more broadly faced, also considering potential underuse phenomena. Indeed, several studies highlighted that this is a frontier issue since guidance is being designed to help practitioners to use biologics appropriately in the treatment of psoriasis across the world [40,41,42]. More studies are needed to understand the economic impact of adopting mAbs for large psoriatic cohorts.

## 4. Materials and Methods

### 4.1. Study Setting

#### 4.1.1. The Italian Healthcare System

The Italian healthcare system is a universal decentralized Beveridge system that includes nineteen regions and two autonomous provinces (APs) [43]. The healthcare system is currently organized and administered at three levels: national, regional, and local. The national government is in charge of identifying core health benefits that should be equally granted across the country and distributing economic resources to the regions via general taxation. The regional governments oversee, organize and deliver primary, secondary, and tertiary healthcare services, as well as pharmaceuticals, preventive, and health promotion services. Specifically, they define their own regional health plans, allocate the budget within their systems and monitor the quality, appropriateness, and efficiency of the services provided. The local level ensures the provision of primary, secondary, and tertiary healthcare services, as well as preventive and health promotion services via Local Health Authorities (LHAs), public hospital institutions and recognized private hospitals.

The study was conducted in Tuscany (population of approximately 3,700,000 inhabitants)—a region of central Italy—in the period between 1 January 2016 and 31 December 2020.

The Tuscan Regional Healthcare System is organized in three LHAs: North-West LHA, Centre LHA, and South-East LHA: the LHAs are geographically based organizations that are responsible for delivering public health services, community healthcare services, and primary care directly. Secondary and specialist care is provided through directly managed facilities or by outsourcing to public hospital institutions or private accredited providers. There are four Teaching Hospitals (TH): Pisa TH, Siena TH, Careggi TH, Meyer TH, and three Area Vasta (AV): North-West AV, Centre AV, and South-East AV. The entity is appointed to coordinate Local Health Authorities and Teaching Hospitals.

#### 4.1.2. The IRPES (Italian Regional Performance Evaluation System)

Tuscany, together with ten other Italian region/autonomous provinces, voluntary joined the IRPES, a Performance Measurement System developed in 2004 by the Management and Health Laboratory (Mes-Lab) of Scuola Superiore Sant’Anna together with the Tuscan Regional Healthcare System. The IRPES is populated by more than 300 indicators measuring and evaluating several performance dimensions (from financial performance to patient satisfaction) of public healthcare organizations. The regional health systems that join the IRPES collaborate with each other and with the Mes-Lab research group for the definitions of the indicators and their calculation methodology and to conduct an in-depth analysis of administrative data. Each regional health system is then in charge of processing its own data, and all the output produced is part of a shared discussion process. The distinctive feature of IRPES is that it is used by the regional systems to produce regulations, and define targets and priorities of their health systems [38].

### 4.2. Study Drugs

Among the selected mAbs, the following were prescribed for psoriasis treatment in the Tuscany region during the study period: Adalimumab, Certolizumab, Etanercept, Guselkumab, Infliximab, Ixekizumab, Sekukinumab, Ustekinumab, Apremilast, Brodalumab, Risankizumab, Tildrakizumab.

The appropriate prescription was evaluated according to drug compendia [19,20], international and national guidelines [23,24]. At the beginning, an analysis of the timing and the specificity of their prescribing was carried out. The selected mAbs were recommended to be prescribed as second-line treatment for psoriasis. Second line-treatment is defined as the ‘*Treatment that is given when initial treatment (first-line treatment) doesn’t work or stops working*’ [44]. The list of pharmaceuticals that the Tuscan Regional Healthcare System, according to guidelines, recommend administering as first-line treatment were Acitretin, Cyclosporin, Dimethylfumarate and Methotrexate, as per Table 4 below. The supply of mAbs as first-line treatment is accordingly considered as potentially inappropriate in this research.

### 4.3. Data Source

Data were extracted from large administrative databases systematically collected by the Tuscan Regional Healthcare System. The analysis was performed on four administrative databases: inpatient databases—SDO (Schede di Dimissione Ospedaliera), outpatient data SPA (Prestazioni Ambulatoriali), and drugs data: FED (Farmaci erogati direttamente dalle strutture) and SPF (Prestazioni farmaceutiche in regime convenzionale). SDO database collects information on all episodes of hospitalization in public and private accredited hospitals in the region. SPA database collects all specialist healthcare outpatient services provided both in public and private accredited facilities. FED database collects drugs delivered directly from public facilities operating in the region, drugs distributed on behalf of public facilities through local pharmacies affiliated, drugs for haemophiliacs administered to hospitalized patients, drugs provided in the course of outpatient services and cancer drugs administered to patients in Day Hospital (DH) or outpatient settings. 

SPF database records the drugs supplied by pharmacies, both public and private, in the regional territory. The medicines dispensed are charged to the INHS and require the user provides the pharmacist with a medical prescription. Pharmaceutical databases contain the financial amounts paid by the Health System for the supply of medicines. All the databases link services or product provided (i.e., hospitalizations, outpatient services and drugs) to a patient ID. The patient ID allows for record linkage between the different administrative databases. This identifier prevents the patient’s identity and other sensitive data from being traced. The study was carried out in compliance with Italian law on privacy, and approval by an Ethics Committee was not required.

### 4.4. Data Analysis

The cohort of psoriatic patients were firstly identified through SDO and SPA databases. Specifically, all the patients that were either hospitalized for psoriasis reasons according to a list of International Classification of Diseases codes ICD9-CM (6960, psoriatic arthropathy; 6962, parapsoriasis; 6961, 6998, and other psoriasis) in 2020 or received an outpatient visit with a final psoriasis diagnosis in 2020 were identified as the study cohort of interest. The SPA database in Tuscany contains the final diagnosis since 2018. Exemption codes were not used since they were not available in the administrative datasets analyzed. No age limits were set. Through a system of record linkage, the cohort was retrospectively followed for the period 2016–2020 across the FED and SPF databases to identify the first and the last pharmaceutical supply that occurred in the selected five years. All the supplies that occurred before 2016 were not identified due to dataset unavailability. Thus, the 2016 or later supply was considered the first one. As a first step, we checked the drug the cohort received as the first and last supply and the respective line of treatment. As a second step, the number of patients in the cohort treated in 2020 with a second-line was investigated for each of the above-mentioned mAbs. As a final step, for patients treated in 2020 with one of the selected mAbs, we verified that the percentage of those that started the treatment with a first-line or with a second-line drug different from the one of 2020 (switch within the same line of treatment, but across molecules) or with the same drug (the patient received the same mAb as first and last supply). Figure 2 illustrates the four steps of the analysis. The patients that started the treatment with a second-line drug were regarded as potentially inappropriate treated, even though the analysis considered only five years of treatment and psoriasis could have been diagnosed and treated previously. All the analyses were conducted both at the regional and Area Vasta (AV) levels. Data were processed using SAS^®^, version 9.4.

## 5. Conclusions

The study shows that in the Tuscany Region, the potentially inappropriate prescribing of mAbs is quite relevant: thus, it has to be considered a matter the Tuscan Regional Healthcare System has to bring attention to. Our research represents a first step to raise awareness of the urgency of improving clinical appropriateness in order to guarantee equity for the population and sustainability for the healthcare system. Indeed, mAbs are particularly costly for the healthcare system and present an inappropriate prescription drain of several economic resources that could be better allocated elsewhere. It is recognized that PES could be a valuable tool to support healthcare systems in improving performances (i.e., clinical, economic), thus also in monitoring innovative drugs’, such as mAbs, appropriate prescription.

## Figures and Tables

**Figure 1 pharmaceuticals-15-01238-f001:**
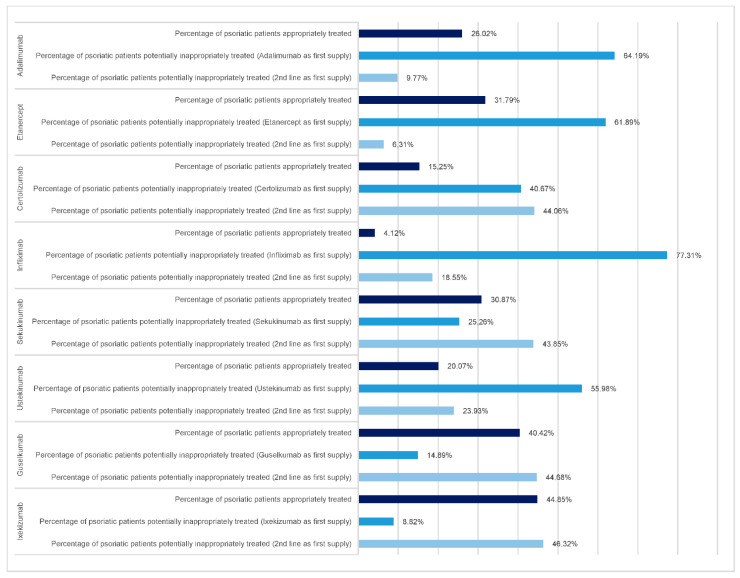
This figure reports the percentage of both appropriately and potentially inappropriately treated 2020 psoriatic patients for each mAb in Tuscany.

**Figure 2 pharmaceuticals-15-01238-f002:**
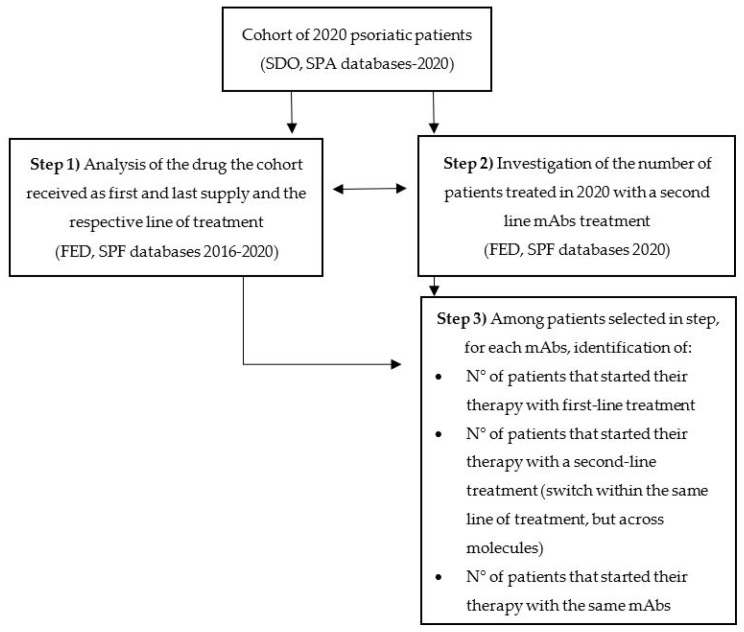
The figure shows the different steps of the analysis.

**Table 1 pharmaceuticals-15-01238-t001:** This table shows the line of treatment the cohort of patients started and ended with (either first or second-line).

First Supply, Line of Treatment	Last Supply, Line of Treatment	N° of Patients	% Over the Total Cohort
1st	1st	3191	55.56%
1st	2nd	608	10.58%
2nd	1st	151	2.63%
2nd	2nd	1793	31.22%
Cohort of 2020 psoriatic patients		5743	

**Table 2 pharmaceuticals-15-01238-t002:** This table shows the number of psoriatic cohort patients treated in 2020 with a second-line mAbs.

Tuscany Region	Number of Patients Treated in 2020 with mAbs	Cohort of 2020 Psoriatic Patients	% Over the Total Cohort
Adalimumab	634	5743	11.04%
Etanercept	412	5743	7.17%
Sekukinumab	285	5743	4.96%
Ustekinumab	259	5743	4.50%
Ixekizumab	136	5743	2.37%
Infliximab	97	5743	1.69%
Guselkumab	94	5743	1.64%
Certolizumab	59	5743	1.03%
Total	1976	5743	

**Table 3 pharmaceuticals-15-01238-t003:** This table shows the average cost per supply and standard deviation for molecules considered in the analysis.

Molecule	Average Cost per Supply	SD
Acitretin	9.15 €	±1.27 €
Cyclosporin	38.94 €	±23.35 €
Dimeltifumarate	937.57 €	±0.18 €
Methotrexate	10.06 €	±3.05 €
Adalimumab	131.33 €	±0.005 €
Etanercept	299.20 €	±0.005 €
Sekukinumab	948.66 €	±0.18 €
Ustekinumab	2589.3 €	±0.01 €
Ixekizumab	1671.02 €	±0.07 €
Infliximab	318.70 €	±166.42 €
Guselkumab	1815.42 €	-
Certolizumab	734.85 €	±2.45 €

**Table 4 pharmaceuticals-15-01238-t004:** This table reports the line of treatment each drug for psoriasis should be preferably administered to consider the prescription as appropriate.

List of Drugs	Guidelines Recommendation for Appropriate Prescription
Acitretin	1st line treatment
Cyclosporin	1st line treatment
Dimeltifumarate	1st line treatment
Methotrexate	1st line treatment
Adalimumab	2nd line treatment
Certolizumab	2nd line treatment
Etanercept	2nd line treatment
Guselkumab	2nd line treatment
Infliximab	2nd line treatment
Ixekizumab	2nd line treatment
Sekukinumab	2nd line treatment
Ustekinumab	2nd line treatment
Apremilast	2nd line treatment
Brodalumab	2nd line treatment
Risankizumab	2nd line treatment
Tildrakizumab	2nd line treatment

## Data Availability

Data is contained within the article and Appendix A.

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
