# Peer review of "Monitoring Appropriate Monoclonal Antibodies Prescribing via Administrative Data: An Application to Psoriasis"

_pharmaceuticals, 2022, doi:10.3390/ph15101238_

Round 1

Reviewer 1 Report

Dear Authors,

This paper shows a pilot study on the prescribing of monoclonal antibodies revealing potential inappropriate prescribing in a region of Italy. The study was an in-depth analysis on large administrative data collected from four databases.

 Comment: In the text that refer to table 4 is not clear why the mention is for Guselkimab and Ixekizumab and not Ustekinumab (has lower SD than Ixekizumab).

Author Response

I would like to thank the reviewer for the notice. Table 4 shows the average cost per supply and standard deviation for molecules considered in the analysis and its aim is to highlight that the inappropriate usage of second-line treatment is particularly costly, independently of the SD. The table is referred to Guselkumab and Ixekizumab in the sense that those molecules when inappropriately prescribed excise significant economic pressure, similar to Ustekinumab. Thus, I would independently refer to one mAbs or the other, independently of the SD that in both case is minimum.

Reviewer 2 Report

This paper presents the analysis of the proscription of monoclonal antibodies in psoriasis treatment in Tuscany, Italy. The analysis was based on the data from databases collected by the regional healthcare system. The results are sound and the methodological quality is appropriate for the publication. I recommend to approve this manuscript for publication.

Author Response

I would like to thank the reviewer. Since the reviewer proposed no modifications, I  would not apply any revision to the original manuscript accordingly.

Reviewer 3 Report

a catchier, "tastier", less neutral title would serve better for the reader

the introduction should be shortened

authors must jump off their comfort zone and that of their knowledge on Italian regulations only, and must put their results into European context (of Psoriasis). Perhaps discuss little further in the "Discussion". 

i also encourage the authors to extend their studies/awareness to other geographical areas (perhaps outside of Italy, too) in psoriasis

English is acceptable

i accept it after minor changes

Author Response

I would love to thank the reviewer for the precious comments.
As suggested the introduction was shortened and the discussion section widened by referring also to the international context. Indeed, the appropriate prescription of mAbs, and drugs in general, in psoriasis is a worldwide topic. 

All the modifies could be seen in track change in the world file attached below.
